# Beyond Single Path Integrated Gradients for Reliable Input Attribution via Randomized Path Sampling

## Abstract

Input attribution is a widely used explanation method for deep neural networks, especially in visual tasks. Among various attribution methods, Integrated Gradients (IG) (Sundararajan et al., 2017) is frequently used because of its model-agnostic applicability and desirable axioms. However, previous work (Smilkov et al., 2017; Kapishnikov et al., 2019; 2021) has shown that such method often produces noisy and unreliable attributions during the integration of the gradients over the path defined in the input space. In this paper, we tackle this issue by estimating the distribution of the possible attributions according to the integrating path selection. We show that such noisy attribution can be reduced by aggregating attributions from the multiple paths instead of using a single path. Inspired by Stick-Breaking Process (Sethuraman, 1991), we suggest a random process to generate rich and various sampling of the gradient integrating path. Using multiple input attributions obtained from randomized path, we propose a novel attribution measure using the distribution of attributions at each input features. We identify proposed method qualitatively show less-noisy and object-aligned attribution and its feasibility through the quantitative evaluations.

## 1 Introduction

Along with the steep improvement and the real world application of the deep learning models (Caruana et al., 2015; Yurtsever et al., 2020), discovering the evidence of the black-box model decision is considered to be important for debugging the malfunction (Lapuschkin et al., 2019) and promise the safety and the fairness (Doshi-Velez & Kim, 2017) of the models. Within the vast literature of explaining the decision of the deep models, input attribution (Simonyan et al., 2013; Bach et al., 2015; Shrikumar et al., 2016; Sundararajan et al., 2017) is one of widely used methods to quantify the relative contribution of each features to the model output. Input attribution provides the explanation in the form of heatmaps, which is useful to indicate the spatial existence of evidences, especially in visual tasks.

Among various approaches to compute the input attributions, Integrated Gradient (IG), one of widely used methods, and its variants (Sundararajan et al., 2017; Pan et al., 2021; Kapishnikov et al., 2021) are of particular interest in our work. These methods explore the input space along the predefined path and integrate the gradients to provide the reliable attributions. The integration path of such methods consists of a baseline which represents the missingness of features and a connecting line between the input and the baseline. With different desired properties, various paths can be used to compute the attribution. For example, Guided IG (Kapishnikov et al., 2021) proposes the adaptive path to alleviate the high and noisy gradients unrelated to the prediction. The selection of baseline can also affect to the attribution results (Štrumbelj & Kononenko, 2014).

While the above methods address the importance of selecting appropriate integration path, in this paper, we claim that the single path is not reliable enough to interpret the decision of neural networks. We provide a simple example that the attribution computed by a single path provides high variance according to different path selection. For better reliability, we propose a novel attribution method to take the expectation of the path-integrated attribution over the distribution of possible paths. To sample from the distribution over the vast variety of possible paths, we adopt the notion of Stick-

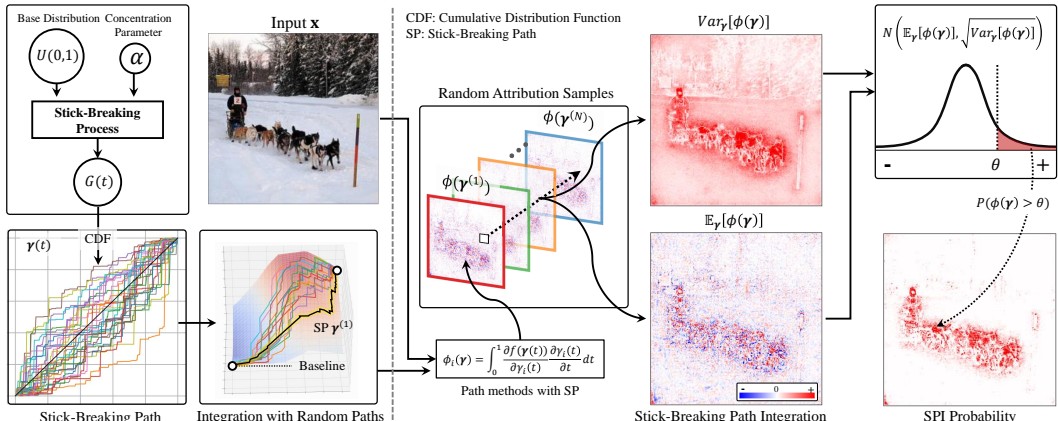

Figure 1: An illustration of Stick-breaking Path Integration (SPI) for the given input **x**. Using the realized distribution $G$ from SBP, we randomly generate the integration path in the input domain by taking CDF of each distribution (colored lines in the left-bottom). From the sampled paths, we apply the gradient integration along each path to gather the multiple attribution samples. By taking the average, the attribution of SPI can be obtained.

Breaking Process, which is one sort of stochastic processes that samples the probability distribution. The main contributions of our work are summarized as,

- Address the inconsistency of attribution according to the selection of the integration path, and propose a novel attribution method that takes the expectation over the distribution of random paths to retain the reliability of attribution.

- Propose a sampling method to generate a random integration path inspired by the Stick-Breaking Process. From the proposed method, we can generate the vast integration paths efficiently.

- Evaluate the attribution in qualitative and quantitative measure to validate the reliability of the proposed method on various of architecture of the networks.

## 2 RELATED WORK

**Attribution methods** The input attribution methods aims to measure the relative sensitivity of the model output with respect to the input features. Saliency method (Simonyan et al., 2013) is a simple approach to use the gradient as attribution. Then Grad*Input method (Shrikumar et al., 2016) is proposed to multiply the input with the gradient for better input alignment. FullGrad (Srinivas & Fleuret, 2019) proposes to use the bias gradient in addition to Grad*Input. Guided Backpropagation (Springenberg et al., 2014) suggests to consider only features that positively contributes to the prediction by ignoring the negative backpropagated gradients. Layerwise Relevance Propagation (LRP) (Bach et al., 2015; Nam et al., 2020) is another method to modify the backward propagation. LRP proposes the relevance propagation rules based on the Taylor decomposition. There exists a family of attribution methods which do not require the access to the internal properties (e.g., gradients, parameters). LIME (Ribeiro et al., 2016) trains a surrogate linear model, which resembles the original model for the data that features are masked out from the input to be explained. RISE (Petsiuk et al., 2018) computes the attribution by aggregating the model outputs from the multiple random masked inputs.

**Path-based attribution methods** Integrated Gradients (IG) (Sundararajan et al., 2017) is one of widely used input attribution method. It is built upon the game-theoretic notion of pay-off distribution method, Aumann-Shapley value (Aumann & Shapley, 2015). IG has several desirable properties, called axioms, which supports the reliability of the attribution. IG is calculated by integrating the gradients along the path from the baseline to the input. Based on this work, several extended research has been performed. To reduce the noise in the attribution, SmoothGrad (Smilkov et al., 2017) takes the average of the multiple random noise added inputs and NoiseGrad (Bykov et al., 2021) inject

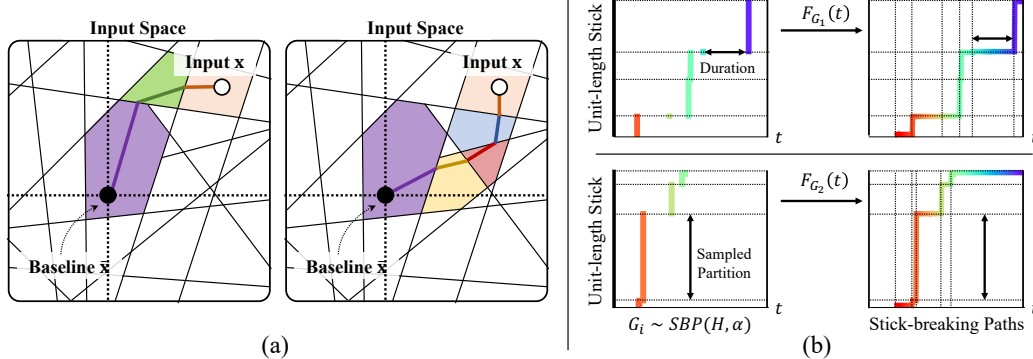

Figure 2: (a) An illustrative example of various path integrated attributions with difference path choices. Each colored region indicates the passed different decision region along the integrated path. We identify that the complexity of integration (e.g., the number of decision regions) can be determined by the shape of the integrated path. (b) An illustrative example of converting the realized partitions of a stick to the integration paths.

noise to the weight parameters. From the observation that the sum of gradient of multiple features is more stable than the individual gradients, XRAI (Kapishnikov et al., 2019) proposes the method to merge the attribution of neighboring features using multi-level superpixels. Some researches try to modify the integration path with their own desired characteristics (Kapishnikov et al., 2021; Pan et al., 2021).

**Evalutation of attribution methods** Even though a plethora of attribution methods are proposed, evaluating the reliability or the correctness are still challenging. Pixel flip is an causal metric between the input and the prediction confidence (Samek et al., 2017). Pixel flip is calculated by setting the value of each feature to zero in order of increasing or decreasing order of attributions. As an extension, insertion and the deletion game is proposed by starting the insertion game from the blurred image to avoid the spurious effects when small number of pixels are inserted (Petsiuk et al., 2018). RemOve-And-Retrain (ROAR) measures the performance drop when the dataset is reorganized by perturbing the input images with the average pixel value in decreasing order of attribution (Hooker et al., 2019). Steep performance drop in ROAR indicates that the attribution method correctly points out important features which are actually used to train and inference.

# 3 STICK-BREAKING PATH INTEGRATION

In this section, we propose our new path-based attribution method, Stick-breaking Path Integration (SPI). We first provide some backgrounds about Integrated Gradients (IG) (Sundararajan et al., 2017) and its family of attribution methods, path methods. From the insight that the attribution highly depends on the path selection, we propose to take the expectation over the distribution of random paths for reliability of attribution. Due to the difficulty in defining the distribution of paths, we propose a sampling process which is motivated by the Stick-Breaking Process (SBP) (Sethuraman, 1991). Using the multiple attributions computed by randomly sampled paths, we finally propose SPI. In addition, we propose a visualization method for SPI, SPI-P, which utilizes the stochastic property of the sampling process.

## 3.1 PATH METHODS AND INTEGRATED GRADIENTS

Integrated Gradient (IG) (Sundararajan et al., 2017) is proposed as an explanation method for neural networks by adopting the Aumann-Shapley value (Aumann & Shapley, 2015), which is a credit allocation method developed for the cooperative game theory. Based on such allocation strategy, IG aims to compute the contribution of input features to the model prediction. For a differentiable model

function $f$, the input $\mathbf{x}$ and the baseline $\bar{\mathbf{x}}$, IG on $i$-th feature is given as

$$\phi_i(\boldsymbol{\gamma}) = \int_0^1 \frac{\partial f(\boldsymbol{\gamma}(t))}{\partial \gamma_i(t)} \frac{\partial \gamma_i(t)}{\partial t} dt, \quad \boldsymbol{\gamma}(t) = \bar{\mathbf{x}} + t(\mathbf{x} - \bar{\mathbf{x}}) \tag{1}$$

where $\boldsymbol{\gamma}(t)$ represents a continuous path from the baseline $\bar{\mathbf{x}}$ to the input $\mathbf{x}$ and $\gamma_i(t)$ refers to the value of $i$-th feature at step $t$ along the path. Previous work has shown that integrating over different path $\boldsymbol{\gamma}$ yields different type of attribution method (Pan et al., 2021; Kapishnikov et al., 2021). We call this group of attribution methods with arbitrary selection of path $\boldsymbol{\gamma}$ as *path methods*. However, the intermediate gradient in the single path is likely to be noisy and it reduces the reliability and interpretability of the obtained attribution. Several work has shown such noise can be reduced by averaging IG over randomly perturbed inputs (Smilkov et al., 2017), randomly perturbed weights (Bykov et al., 2021) or average pooling over spatial location (i.e., local pixels for image) (Kapishnikov et al., 2019).

In this work, we propose to aggregate the attribution from multiple paths with a fixed baseline sampled from the proposed distribution of paths to reduce the noise and improve the confidence of the attribution. We first define the integration path and its desired properties.

**Definition 3.1** (Integration Path). The integration path is a mapping function from $t \in [0, 1]$ to the input domain $\mathcal{X}$, $\boldsymbol{\gamma} : [0, 1] \mapsto \mathcal{X}$. The path should satisfy two properties; (1) the path starts from the baseline, $\boldsymbol{\gamma}(0) = \bar{\mathbf{x}}$, and ends at the input to be attributed, $\boldsymbol{\gamma}(1) = \mathbf{x}$ and (2) the path monotonically proceeds from $\bar{\mathbf{x}}$ to $\mathbf{x}$, i.e., $\frac{d\gamma_i(t)}{dt} = C(x_i - \bar{x}_i)$ for $C \geq 0$.

In the rest of the paper, we denote the bold symbols for the vector (e.g., $\mathbf{x}$). We use the subscript to represent the indexed component (e.g., $x_i$). We use the superscript to represent the different indexed instances (e.g., $\mathbf{w}^{(1)}, \ldots, \mathbf{w}^{(j)}, \ldots$).

## 3.2 RELATIONSHIP BETWEEN PATH AND ATTRIBUTION

A neural network equipped with the partial linear activation, such as ReLU, is known to have the form of the piece-wise linear function (Montufar et al., 2014). The piece-wise linear function is defined by multiple linear functions, where each linear function is only feasible in corresponding linear region, $\mathcal{R}^{(j)}$. Such piece-wise linear function $f$ can be formulated as follow with each linear region $\mathcal{R}^{(j)}$,

$$f(\mathbf{x}) = \begin{cases} \mathbf{w}^{(1)T}\mathbf{x} + \mathbf{b}^{(1)} & \mathbf{x} \in \mathcal{R}^{(1)} \\ \ldots \\ \mathbf{w}^{(j)T}\mathbf{x} + \mathbf{b}^{(j)} & \mathbf{x} \in \mathcal{R}^{(j)} \\ \ldots \end{cases} \tag{2}$$

Then the path method provides the attribution in terms of weighted sum of corresponding weight vectors in each region,

$$\phi_i(\boldsymbol{\gamma}) = \sum_j \frac{w_i^{(j)} \alpha_i^{(j)}(\boldsymbol{\gamma})}{\sum_{j'} \alpha_i^{(j')}(\boldsymbol{\gamma})}, \quad \alpha_i^{(j)}(\boldsymbol{\gamma}) = \int_{t=0}^1 \frac{d\gamma_i(t)}{dt} \delta^{(j)}(\boldsymbol{\gamma}(t)) dt \tag{3}$$

where $\alpha_i^{(j)}(\boldsymbol{\gamma})$ represents the projected length of the path $\boldsymbol{\gamma}$ to the $i$-th feature that passes through the region $\mathcal{R}^{(j)}$. The delta function $\delta^{(j)}(\boldsymbol{\gamma}(t))$ returns 1 if $\boldsymbol{\gamma}(t) \in \mathcal{R}^{(j)}$ and otherwise 0. Even though $\phi_j$ in Equation 3 sums over every region, we note that it is equivalent to the summation over the regions that path $\boldsymbol{\gamma}$ passes through, because $\alpha_i^{(j)}(\boldsymbol{\gamma}) = 0$ for the region $\mathcal{R}^{(j)}$ that $\boldsymbol{\gamma}$ does not pass through.

It has been shown that to increase the expressivity of the DNNs, the number of linear regions should also increase (Xiong et al., 2020). With the high dimensional input space and the vast number of linear regions, using a single path for the path methods would induce high uncertainty and variance to the resulting attributions. Figure 2a illustrates that even in 2-dimensional space with several linear regions, different selection of path aggregates mostly different weight vectors of corresponding regions. To alleviate such uncertainty with selecting an integration path, we propose to compute the expectation of path method over the distribution of possible paths as follow,

$$\mathbb{E}_{\boldsymbol{\gamma}}[\phi_i(\boldsymbol{\gamma})] = \int_{\boldsymbol{\gamma}} \int_0^1 \frac{\partial f(\boldsymbol{\gamma}(t))}{\partial \gamma_i(t)} \frac{\partial \gamma_i(t)}{\partial t} dt P(\boldsymbol{\gamma}) d\boldsymbol{\gamma} \tag{4}$$

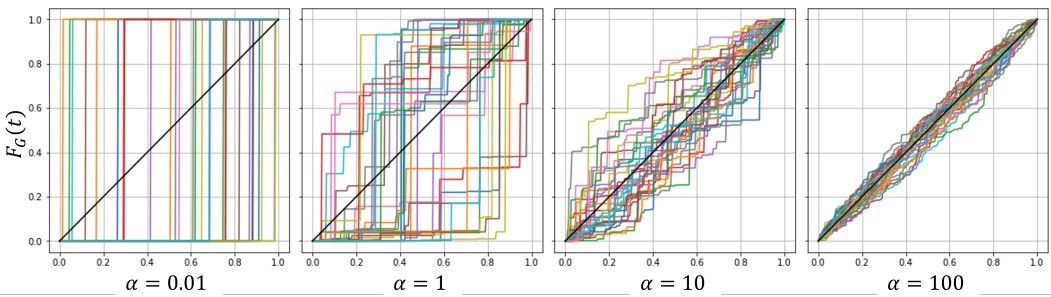

Figure 3: Visualization of CDF of SBP realized distributions $G$ with different concentration parameter $\alpha$ (colored lines). The base distribution is given as uniform distribution, $U(0,1)$, and its CDF is given as the straight line (black line). The realized CDF converges to the CDF of the base distribution when the concentration hyperparameter $\alpha$ increase.

### 3.3 RANDOMIZED PATH SAMPLING

As the distribution of the path, $P(\boldsymbol{\gamma})$, is difficult to be defined, we propose an alternative approach to randomly generate the path with the variety. When generating a path, we have two choices to select; (1) whether proceed to the destination or not and (2) how far to proceed. We reduce this problem to the Stick-Breaking Process (SBP) (Sethuraman, 1991). SBP is a generative process to obtain random fractions of a stick with its initial length of 1. The fraction obtained by SBP can be regarded as how much portion from the baseline to the input should move at each step. In this configuration, the number of fraction can be regarded as the number of steps in the path and each length of fractions can be regarded as each length of steps. Figure 2b depicts how the sampled partitions of the unit length stick can be used to build the integration path. Each sampled fraction is represented a Probability Mass Function (PMF), and such PMF is defined as follow,

$$G(t)=\sum_{k=1}^{\infty} \pi_k \delta_{t_k}(t) \sim SBP(H,\alpha), \quad \pi_k=\beta_k\prod_{i=1}^{k-1}(1-\beta_i), \quad \beta_k \sim Beta(1,\alpha), \quad t_k \sim H \quad (5)$$

where $\delta_{t_k}(t)$ is a delta function that returns zero everywhere, except for $\delta_{t_k}(t=t_k)=1$ and $H$ is a base distribution. The expectation of $G$ sampled by SBP is desired to estimate the base distribution $H$. The hyperparameter $\alpha$, also known as concentration parameter, controls the realized distribution $G$ to be more similar to the base distribution if $\alpha$ takes larger value.

From the sampled PMF $G(t)$, we define the CDF $F_G(t)$ which can be used to build the integration path function. Let the integration path be

$$\gamma_i(t) = \bar{x}_i + F_G(t)(x_i - \bar{x}_i). \quad (6)$$

We note that the Equation 7 satisfies the properties of Definition 3.1; (1) $\gamma_i(0)=\bar{x}_i$ and $\gamma_i(1)=x_i$ because $F_G(0) = 0$ and $F_G(1) = 1$, and (2) $dF_G(t)/dt = G(t) \geq 0$. With taking the base distribution $H$ to be the uniform distribution, $H = U(0,1)$, whose CDF is equivalent to the IG straight path. Figure 3 depicts the CDF of SBP realized distributions using different $\alpha$. When the value of $\alpha$ increases, the realized CDF converges to the CDF of $U(0,1)$ (black line), and we call this CDF as the base path.

**Definition 3.2** (Stick-breaking Path (SP)). Given a hyperparameter $\alpha_i > 0$, the Stick-breaking Path (SP) of $i$-th feature from the baseline $\bar{x}_i$ to the input $x_i$ is defined as the multiplication of the CDF of the distribution $G_i$ obtained by SBP and the difference between $\bar{x}_i$ and $x_i$,

$$\gamma_i(t;\alpha_i) = \bar{x}_i + F_{G_i}(t)(x_i - \bar{x}_i), \quad G_i \sim SBP(U(0,1),\alpha) \quad (7)$$

With the sampling process of SP and hyperparameter $\alpha$, we finally propose a new attribution method, Stick-breaking Path Integration (SPI).

**Definition 3.3** (Stick-breaking Path Integration). For a model $f$ and a baseline $\bar{\mathbf{x}}$, SPI is defined as the expectation of the path method over the path distribution,

$$SPI_i(\mathbf{x};\alpha) = \mathbb{E}_{\boldsymbol{\gamma}}[\phi_i(\boldsymbol{\gamma})] = \mathbb{E}_{\mathbf{G}}\left[(x_i - \bar{x}_i)\int_0^1 \frac{\partial f(\bar{\mathbf{x}} + F_{\mathbf{G}}(t) \odot (\mathbf{x} - \bar{\mathbf{x}}))}{\partial(\bar{x}_i + F_{G_i}(t)(x_i - \bar{x}_i))}\frac{dF_{G_i}(t)}{dt}dt\right] \quad (8)$$

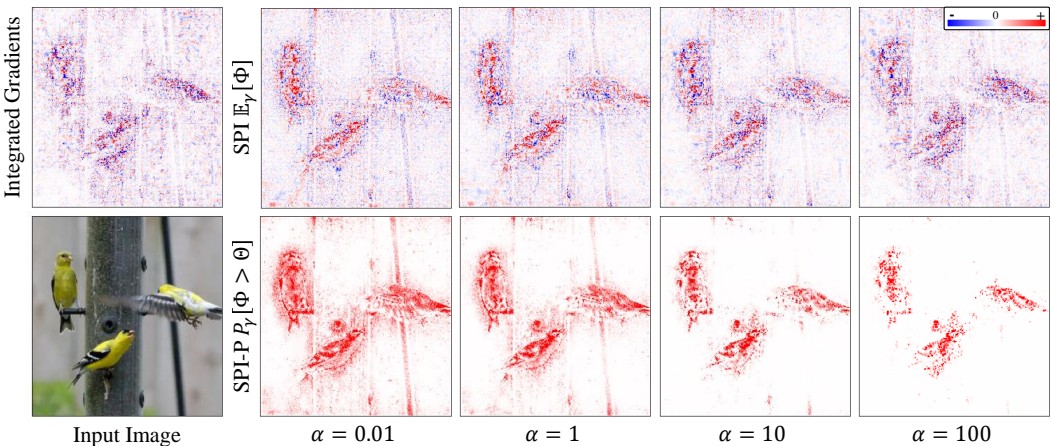

Figure 4: Input attribution obtained by proposed SPI method with varying the hyperparameter $\alpha$. With increasing $\alpha$, SPI-E becomes noisy and resembles the attribution of IG. With decreasing $\alpha$, the noise is reduced on the objects, but the checkered pattern appear in the background of the image. SPI-P with low $\alpha$ also returns high probability for the background.

where $\mathbf{G}$ is a vector with each index follows SBP, $G_i \sim SBP(U(0, 1), \alpha)$ and $F_{\mathbf{G}} : [0, 1] \mapsto \mathbb{R}^d$ is a stack of SP that returns a vector in the input space according to the current step $t$.

### 3.4 VISUALIZATION WITH SCORE BASED ON STATISTICS OF ATTRIBUTIONS

As we sample SP from the random process SBP, the attribution computed using SP, $\phi_i(\boldsymbol{\gamma})$, also can be regarded as random variables. Assuming that $\phi_i(\boldsymbol{\gamma})$ follows the Gaussian distribution, the estimated Gaussian distribution of $\phi_i(\boldsymbol{\gamma})$ is given as follow

$$\phi_i(\boldsymbol{\gamma}) \sim N(\mu, \sigma), \quad \mu = \frac{1}{N}\sum_{k=1}^{N}\phi_i(\boldsymbol{\gamma}^{(k)}), \quad \sigma = \frac{1}{N}\sum_{k=1}^{N}\left(\phi_i(\boldsymbol{\gamma}^{(k)}) - \mathbb{E}_{\boldsymbol{\gamma}^{(k)}}[\phi_i(\boldsymbol{\gamma}^{(k)})]\right)^2. \quad (9)$$

We observe that the input attributions obtained by integrating gradient over each randomly sampled path deviates in their own distribution. We categorize such attributions to three; (1) the one with low variance and low mean, (2) the one with low variance with high mean and (3) the one with high variance. For the first and the second cases, we can identify that those features are consistently show low/high contribution with less dependent to the path. However, for the third case, we need to measure how probable that such feature would contribute much.

With this insights, we propose an additional method by assigning the probability of attribution at each feature is sufficiently positive along SBP. Assuming that the attributions for each feature are distributed normally, we propose SPI-P, which measures the probability of attribution to be larger than some threshold $\Theta$. We use the top 5% quantile value of $SPI(\mathbf{x}; \alpha)$ as the threshold.

**Definition 3.4** (SPI Probability (SPI-P)). Assume that the attribution at $i$-th feature follows the Gaussian distribution with estimated $\mu$ and $\sigma$ in Equation 9. Then SPI-P is defined by the CDF of the Gaussian distribution,

$$SPI\text{-}P_i(\mathbf{x}; \alpha) = P(\Phi_i > \Theta) = 1 - \frac{1}{\sigma\sqrt{2\pi}}\int_{-\infty}^{\Theta}\exp\left(-\frac{(\theta - \mu)^2}{\sigma^2}\right)d\theta \quad (10)$$

### 3.5 ANALYSIS ON $\alpha$

We note that $\alpha$ controls the variance of sampled paths in Section 3.3. In this section, we provide the analysis on how the attribution obtained by proposed method differs according to the value of $\alpha$. Figure 4 shows the qualitative comparison of the proposed two methods, SPI-E and SPI-P. The first row in Figure 4 indicates that as $\alpha$ increases, the attribution becomes similar to the attribution of

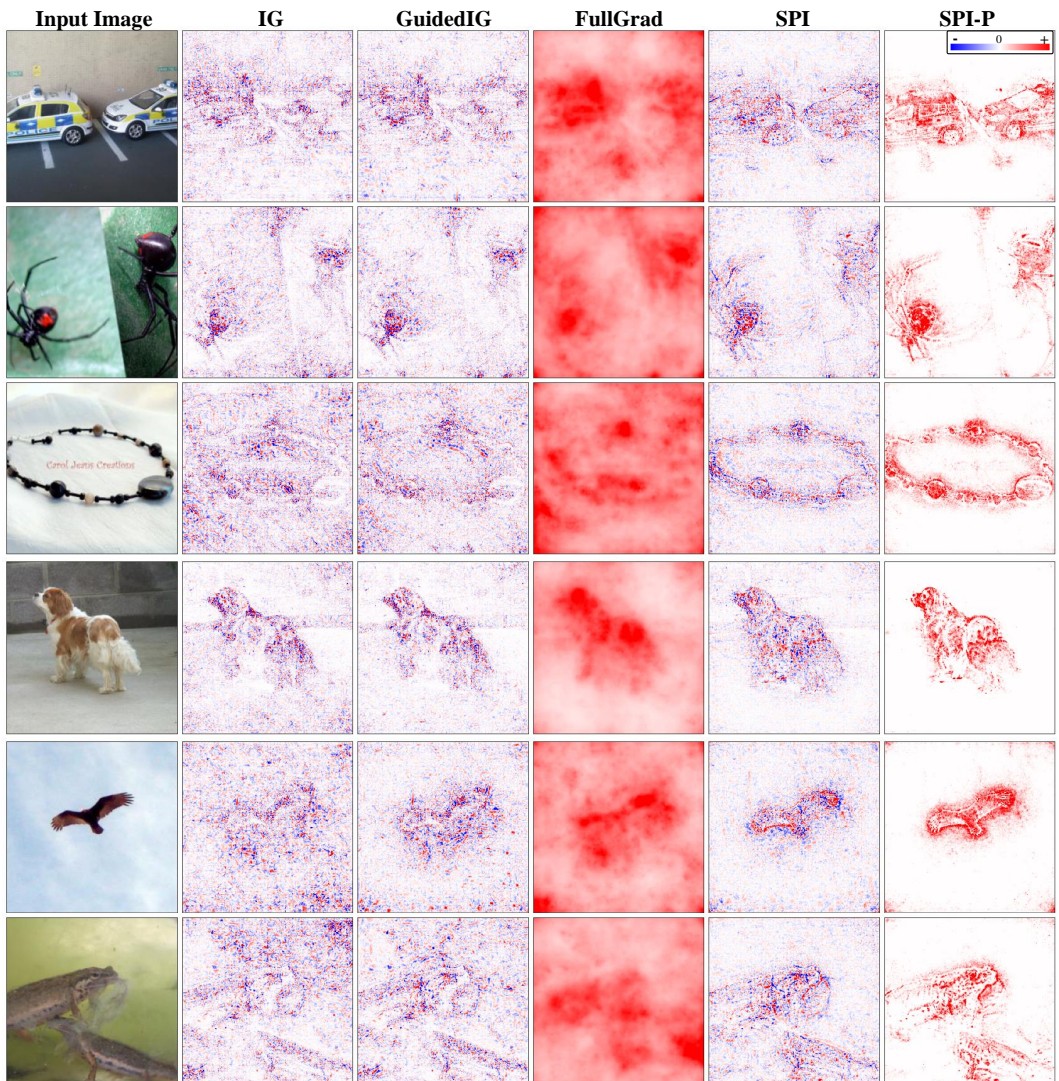

Figure 5: Qualitative comparison among various attribution methods for VGG-16 in the validation dataset of ImageNet. Each column describes the heatmaps obtained by each attribution methods. SPI generates more object-oriented and less noisy attribution heatmaps. With SPI-P, the heatmaps are more distinguishable between the important features and the irrelevant background.

IG. As previously described, if $\alpha$ increase, the realized paths converges to the base path, which is the straight line of IG. The attribution with with high $\alpha$ shows noisy results that the positive and the negative values are alternatively placed nearby. Such noisy attribution has been raised as a problem in gradient-based attribution methods (Kapishnikov et al., 2021). In contrast, the result with low $\alpha$ shows loss noisy attribution. Comparing the attribution from the high and the low $\alpha$, With low $\alpha$, the attribution shows less noisy and more consistent on the object.

## 4    EXPERIMENTS AND RESULTS

In this section, we verify the effectiveness of SPI by the qualitative analysis and the quantitative comparison. We first provide the qualitative comparison among different attribution methods, by providing the attribution heatmaps on the randomly chosen examples. We note that the quantitative evaluation on the attribution methods is challenging because there is no ground truth of the attribution and the ground truth even changes if we train a new model. To alleviate the absence of

Table 1: Insertion/Deletion game of various attribution methods on three models.

| | Insertion (↑ is better) | | | Deletion (↓ is better) | | |
| | VGG-16 | Inception-v3 | ResNet-18 | VGG-16 | Inception-v3 | ResNet-18 |
|---|---|---|---|---|---|---|
| Grad*Input | 0.078 | 0.171 | 0.114 | 0.045 | 0.105 | 0.050 |
| GuidedBProp | 0.094 | 0.145 | 0.124 | 0.113 | 0.162 | 0.145 |
| IG | 0.096 | 0.243 | 0.158 | 0.036 | 0.066 | 0.038 |
| FullGrad | 0.415 | 0.558 | 0.448 | 0.110 | 0.175 | 0.131 |
| GuidedIG | 0.110 | 0.255 | 0.185 | 0.029 | 0.061 | 0.02 |
| SPI (Ours) | **0.443** | **0.704** | **0.515** | **0.019** | **0.051** | **0.018** |

the ground truths, we provide the quantitative comparison using two widely used metrics: (1) pixel insertion/deletion game (Samek et al., 2017; Petsiuk et al., 2018), and (2) RemOve-And-Retrain (ROAR) (Hooker et al., 2019) to identify that the proposed method can assign attributions which are more relevant to model behavior. We select various gradient-based attribution methods for the comparison: Gradient*Input (Shrikumar et al., 2016), Guided BackPropagation (GuidedBProp) (Springenberg et al., 2014), Integrated Gradients (IG) (Sundararajan et al., 2017), FullGrad (Srinivas & Fleuret, 2019), and GuidedIG (Kapishnikov et al., 2021).

## 4.1 QUALITATIVE COMPARISON

For the visual inspection, which is the main application of the input attribution, we qualitatively compare the attribution heatmaps obtained by various methods. For the heatmap visualization, we take the color range to be bounded by top 5% absolute value of each attribution. Figure 5 provides the attribution heatmaps of randomly selected images from the validation set of ImageNet with the pre-trained VGG-16. We identify that the attributions obtained by SPI show more aligned to the target class object and less attributed to the unrelated background. For example, in the third row, SPI correctly focuses on the necklace, while other methods are distracted by the irrelevant pixels in the background. With SPI-P visualization, we also identify that the attribution better isolates the important features from the input image.

## 4.2 PIXEL INSERTION AND DELETION GAME

Pixel flip is first proposed to benchmark if attribution methods correctly capture the relevance between the input features and the model output (Samek et al., 2017). It is later introduced as evaluating the effect on the model prediction caused by the pixel perturbation (Petsiuk et al., 2018). To quantify the relevance between the input features and the model output, pixel flip modifies the pixel values in order of relevance obtained by attribution methods, from high to low. Then it measures the change of softmax output for the target class with the perturbation. Pixel flip consists of two evaluations, the insertion game and the deletion game. The insertion game inserts the original pixel value to the predefined baseline, and the deletion game deletes the original pixel value in the input. If the input attribution is correctly related to the model prediction, then the insertion game would give high score by steep increase of output with the insertion of highly attributed pixels. In the same manner, the deletion game would give low score with correctly related attribution. When modifying the pixel, several choices arise which value to replace. We follow the configuration of previous work (Petsiuk et al., 2018), which use the blurred input for the insertion and the zero value for the deletion.

We use 50k images of the validation set provided by ImageNet (Russakovsky et al., 2015). We use three publicly available pre-trained models: VGG-16 (Simonyan & Zisserman, 2015), Inception-v3 (Szegedy et al., 2016), ResNet-18 (He et al., 2016). Table 1 indicates the insertion and the deletion results for the various attribution methods and model architectures. We identify that SPI shows the best performance in both games on entire three architectures.

## 4.3 REMOVE-AND-RETRAIN (ROAR)

ROAR (Hooker et al., 2019) is another metric to evaluate if the attribution method correctly indicates the features that are important in the perspective of the model training. ROAR is calculated by

Table 2: ROAR evaluation of various attribution methods on ResNet-18 trained with CIFAR-10.

| Removed % | 10 | 30 | 50 | 70 | 90 |
|---|---|---|---|---|---|
| Grad*Input | 52.76±0.95% | 39.74±1.01% | 33.22±1.33% | 29.25±0.43% | 24.92±1.39% |
| GuidedBProp | 67.51±0.40% | 63.70±0.80% | 61.91±0.85% | 59.99±0.93% | 52.73±0.52% |
| IG | 39.70±0.79% | 26.23±1.11% | 21.46±0.53% | 17.73±0.58% | 15.73±0.82% |
| FullGrad | 67.49±0.79% | 56.84±0.97% | 42.31±0.75% | 26.10±1.19% | 14.82±1.09% |
| GuidedIG | 38.90±1.63% | 24.23±1.01% | 20.64±1.22% | 18.11±0.75% | 15.66±0.81% |
| **SPI (Ours)** | **31.00±0.48%** | **19.70±0.40%** | **16.05±0.91%** | **13.09±0.72%** | **10.88±1.06%** |

measuring the performance drop when the model is re-trained with modified training data. Each sample in the training dataset is modified by removing pixels with top $k\%$ attribution and replacing them with the average pixel value of the input. For the ROAR experiment, we use ResNet-18 architecture and train the model on 50k images of training set provided by CIFAR-10 dataset (Krizhevsky et al., 2009). For the training, we use the Adam optimizer (Kingma & Ba, 2014) with learning rate 3e-4 and 100 epochs. After training with each modified dataset, the performance of the trained model is measured with the standard test dataset with 10k images in CIFAR-10. We note that the attribution method captures more relevant features if the test accuracy is lower. Table 2 shows the test accuracy measure in the ROAR experiment for each attribution method. We repeatably conduct 10 trials of the experiments, where the parameters are random initialized at each trial and fixed between attribution methods. Table 2 shows the test accuracy measure in the ROAR experiment for each attribution method. The result indicates that the model trained on the modified dataset with SPI steeply decreases the test accuracy even with 10% removed. We conclude that SPI is effective in identifying the input features which are important to train the DNNs models.

## 5 DISCUSSION

In this paper, we proposed the novel attribution method, Stick-breaking Path Integration (SPI), to provide more reliable explanation on the relationship between the input features and the model decision. Based on the path method, which is computed by integrating the gradients along the path in the input space, we first provide the affect of path selection to the computation of the input attribution. By raising the necessity of considering multiple paths for the reliable attribution, we propose to average the path-based attribution over the distribution of paths. We qualitatively show that our method provides attributions with more object-aligned and less noisy. We also provide the quantitative evaluations, pixel flip and ROAR, to identify that our method is well-aligned with the model behavior.

Our work also sheds light for several future works that would provide more reliable or meaningful attributions. The selection of the base distribution would be different to the uniform distribution. For example, one may use the Gaussian distribution to control the path to proceed at early steps or late steps by managing the mean of the distribution. Using different path sampling method would be another approach. Our method assumes the paths are distributed uniformly, but one may define a distribution with several modes. Randomizing the baseline would also diversify the sampled integration path. We expect our work would provide new aspect of attribution methods to inspect the black-box models.

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
