# OpenReview forum: "Beyond Single Path Integrated Gradients for Reliable Input Attribution via Randomized Path Sampling"
_ICLR.cc/2023/Conference — Submitted to ICLR 2023_

### Official Review · Reviewer_yFiz · 2022-10-18

**Confidence:** 4
**Clarity, Quality, Novelty And Reproducibility:** 1. Some of the important proof is mis…
**Correctness:** 2
**Technical Novelty And Significance:** 2
**Empirical Novelty And Significance:** Not applicable
**Recommendation:** 6

**Strength And Weaknesses:**

Weaknesses:

1. Given the successful work of IG method, this submission extends the attribution map integration from single path to distributed multiple paths, which is a little bit incremental.

2. There are some incomplete sentences and typos in the submission. For example, 'Sanity checks on the attribution is an'.

3. For equation(2), ReLU, which is the most commonly used activation in vision, only with two linear regions, thus the proposed path choosing method may not important. Moreover, the discussion about the non-linear activations, such as sigmoid, should be discussed.

4. When the value of the hyperparameter /alpha is small, how will the path choosing process be affected? With a less constrained random choosing? Or pushed far away from the base distribution H?

5. Eq(9) and Eq(10) based on the assumption that the \phi follows the Gaussian Distribution. Therefore, the proof of the distribution of \phi is a Guassian should be involved.

**Summary Of The Paper:**

The authors propose to reduce the attribution nioses in integrated gradients method by utilizing multiple intergrating paths. Specifically, the Stick-Breaking Process is utilized for aggregating the attribution map from large number of random choosing paths.

**Summary Of The Review:**

The score of this paper can be changed according to the rebuttal response.

---

> ### Comment · Reviewer_yFiz · 2022-11-25
> **Thanks**
>
> Thanks for the response, I have updated my score from 5 to 6.

---

### Official Review · Reviewer_jggk · 2022-10-20

**Confidence:** 4
**Correctness:** 3
**Technical Novelty And Significance:** 1
**Empirical Novelty And Significance:** 2
**Recommendation:** 5

**Clarity, Quality, Novelty And Reproducibility:**

*** Clarity
There are quite a lot of minor grammatical mistakes. Together with the many acronyms, it makes reading the text quite cumbersome at times.

*** Quality
The conduction of experiments seems fine.

*** Novelty
The contribution is novel but not very significant since it is a minor modification of the existing Integrated Gradients explanation method.

*** Reproducibility
There is no accompanying code nor an appendix stating details of the implementations. The method itself is likely relatively easy to implement, though.

**Strength And Weaknesses:**

*** Strengths
Attribution maps are less noisy and show good performance on the evaluated metrics.


*** Weaknesses
My main concerns are with respect to the significance and novelty of the contribution. There are already many attribution methods out there, so I think the minor extension of an already existing method might not be significant enough to be published at ICLR. There is little theoretical motivation/analysis/guarantees for the proposed methods. Comparison to other popular methods (LIME, SmoothGrad) is missing.

**Summary Of The Paper:**

The authors propose the attribution method ``Stick-breaking Path Integration'' which averages the attribution maps obtained by integrating over several different paths using the Integrated Gradient attribution method. They show superior performance qualitatively and quantitatively (pixel flipping, ROAR) compared to other explanation methods (Gradient*Input, GuidedBackProp, IntegratedGradients, FullGrad, GuidedIG).

**Summary Of The Review:**

While the results show superior performance on quantitative metrics, the overall contribution does not seem very significant. I cannot recommend publication at ICLR.

---

### Official Review · Reviewer_tTr4 · 2022-10-24

**Confidence:** 4
**Correctness:** 3
**Technical Novelty And Significance:** 3
**Empirical Novelty And Significance:** 3
**Recommendation:** 6

**Clarity, Quality, Novelty And Reproducibility:**

The paper is clear, easily to read. In general, IGs average the gradient of multiple 'inputs', either over brightness level interpolations or in a local neighborhood. (novelty) Here the authors are proposing to 'keep the same input', and generate different/random estimates (by a StickBreaking-like Process).


**Strength And Weaknesses:**

**Strength**

- Integration of different paths keeping the input fixed
- sampling method to generate a random integration path
- Good results on qualitative examples


**Weaknesses**

- More details in the experiments section would help with comprehension and reading.

For example, the pixel ins/del game. Is the input tiled or is each pixel used as a test unit?

If I understand correctly, given an attribution heatmap, each "unit (=pixel/tile)" is ranked according to the attribution values, in deletion, the highest-ranked 'unit' is replaced with a constant value (=0), the modified input is fed through the network, and the resulting drop in target class score is measured.

- Does the "insertion" test start from a black image?
- Which score is reported in Table 1? The first output or the AUC of the resulting curve?
- If the score is the AUC of the curve, it might be interesting to have a plot of some curves to see the different trends.

The two tasks, del/ins seem exactly complementary. I'm not sure what information is adding the insertion task. On the other hand, it is useful to find the pixels/tiles that  disrupt the output of the network as quickly as possible. Neural networks are sensitive to subtle changes in input, and pixels or tiles don't necessarily have to contain meaningful information to break the network (this is also one of axioms in the IG paper). Therefore, Ancona [1] and others [2,3,4] proposed to reverse the deleting direction, by first removing pixels or tiles ranked as less relevant by the attribution method.

Removing 1) the most important pixels first and 2) the least important pixels first, leads to two different curves, the integral of the two curves provides an estimate of the performance of an attribution method in a quantitative way.





[1] Marco Ancona, Enea Ceolini, Cengiz Oztireli, and Markus Gross. A unified view of gradient-based ¨ attribution methods for deep neural networks. In NIPS 2017-Workshop on Interpreting, Explaining and Visualizing Deep Learning. ETH Zurich, 2017.

[2] Sara Hooker, Dumitru Erhan, Pieter-Jan Kindermans, and Been Kim. Evaluating Feature Importance Estimates. arXiv e-prints, 2018.

[3] Pieter-Jan Kindermans, Kristof T. Schutt, Maximilian Alber, Klaus-Robert Muller, Dumitru Erhan, Been Kim, and Sven Dahne. Learning how to explain neural networks: Patternnet and ¨ patternattribution. In International Conference on Learning Representations, 2018.

[4] Wojciech Samek, Alexander Binder, Gregoire Montavon, Sebastian Lapuschkin, and Klaus-Robert ´ Muller. Evaluating the visualization of what a deep neural network has learned. ¨ IEEE transactions on neural networks and learning systems, 28(11):2660–2673, 2016.




**Summary Of The Paper:**

Attribution methods address the issue of explainability by quantifying the importance of an input feature for a model prediction. In this paper, the authors propose a variant of the integrated gradients (IGs) method through a strategy of "selection of the paths to be integrated / aggregated". The authors propose to generate several possible attribution estimates through a stochastic process, then, they show that attribution is improved by aggregating attributions from multiple paths instead of using a single path.

The main contribution of the paper is a new strategy that allows to improve attribution maps.




**Summary Of The Review:**

I'm satisfied with the response of the authors. The different points I had raised were taken into account and my doubts were clarified.

--

The paper is well written and clear, there is an interesting contribution which consists in extending the IGs method by keeping the input fixed and generating a set of random estimates in order to integrate the different paths and obtain a single attribution map.
The weakness of the paper is in the experimental section which does not contain sufficient details to analyze the results.

---

### Official Review · Reviewer_KHRW · 2022-10-25

**Confidence:** 4
**Correctness:** 3
**Technical Novelty And Significance:** 2
**Empirical Novelty And Significance:** 1
**Recommendation:** 6

**Clarity, Quality, Novelty And Reproducibility:**

Fairly clear, alright quality and some novelty. Results could be hard to reproduce with the stick breaking section is lacking in details and no code is released.

**Strength And Weaknesses:**

IG is a well-established attribution technique and there have been a fair amount of follow-up work to improve it. The authors clearly argue how using a single path can lead to sampling error, and propose a sensible way to sample paths. The results section shows the proposed technique improves on several metrics and is convincing enough. Some critical thoughts follow:

1. The development of section 3.3 is unclear to me. I am also not sure how much the choice of the distribution matters, and this uncommon choice seems unjustified. I would think similar results might be observed by using another discrete distribution (say hypergeometric normalized to a max value of 1), since the CDF is always monotonic and meets the required constraint. I would appreciate authors thought on this and any justification for this specific choice as opposed to a general discussion around any parametrized probability distribution.

2. The result section would benefit from inclusion of smoothgrad as the only other technique that performs some sort of averaging and thus could offer a more competitive baseline.

3. In definition 3.1, the definition of monotonicity is incorrect. It defines the much more constrained linear path. Similar error appears in the description of equation 1 which is constrained to be linear instead of a general path.

4. Many minor editing errors: “we address the single path is not reliable”, “consider the features only increase the value”, “the attribution is an”, “averaging pooling over”, “path is monotonically proceed”, “if \alpha assigned with bigger value”, “with alleviating the absence”, “with DGA” (which is undefined).

5. I also don’t see the clear value of including the visualization method with the proposal of the technique diluting the focus, but this is more subjective.

**Summary Of The Paper:**

In this paper, the authors propose a refinement of Integrated Gradient (IG) by considering several paths from the baseline to the test example. The paths are sampled from a specific discrete distribution, and the results show improvement in some of the established metrics like insertion/deletion tests and ROAR.

**Summary Of The Review:**

After author response:
I am changing my score from 6 to 7. Since that option doesn't exist in the UI (it goes from 6 to 8), I am adding this note.

-------
Overall, the paper makes a sensible change to IG by including path averaging and the results look convincing. On the other hand, the specific path sampling method seems like an over-specification, and the result section glosses over a competitive baseline. In its current draft, I am only moderately inclined for the paper to be accepted at the venue.

---

### Decision · Program_Chairs · 2023-01-20

**Decision:**

Reject

**Justification For Why Not Higher Score:**

The proposed method has a higher computational cost, but did not compare with other existing methods with similar computation budgets.

**Justification For Why Not Lower Score:**

The proposed method is simple and effective compared to the limited baselines (while extensive all have less computation budget).

**Metareview: Summary, Strengths And Weaknesses:**

(a) the paper considers an extension of IG -- to average IG over different paths other than a straight line which are modeled by a e Stick-Breaking Process. They show that the resulting SPI outperforms several existing saliency methods on several benchmarks. (b) the strength of the paper is that they can achieve a set of strong empirical results on several benchmarks comparing with a lot of existing saliency methods (c) The weakness of the paper is that the computational cost for SPI is higher than other averaging baselines (they need to integrate over choices of paths and the paths), while baselines at most average over one quantity (such as IG for the path or SG for the input). This would also limit the impact for SPI in practice, as the reason why most commonly-used methods only average over one quantity is that the computational cost to average over two quantity is quite high.  There exist methods that averages over two quantities (such as IG + SG or IG with random averaging baselines) which are more similar to the proposed SPI in terms of computational cost, but they are not empirically compared. Moreover, many counterfactuals for SPI and IG + SG may overlap, as all data point x'' that are close to the path from x (input) and x'(baseline) may be likely to be sampled by both SPI and IG + SG (or IG + random baselines). Therefore, it is unclear whether averaging over paths is better than averaging over baselines (IG with random baselines) or inputs (IG + SG), and the better performance of SPI may be simply due to the higher computational cost (and averaging more counterfactuals). Other competitive baseline such as RISE are also not compared for the insertion and deletion score, which can be a natural baseline to SPI by limiting the same number of model passes. Due to the higher computational cost of the proposed SPI, a more careful and fair empirical examination is crucial to understand whether the improvement of SPI comes from the computational cost or the design.

**Summary Of Ac-Reviewer Meeting:**

* method is incremental (jggk) -- I wouldn't mind an incremental idea as long as the proposed method is useful enough.
* It's not very novel but it's well done (KHRW) -- Since SPI has higher computational cost than all the compared baselines, I'm not convinced that SPI is sufficiently well done.
* added sufficient comparisons with averaging methods(KHRW) -- Since SPI has higher computational cost, I'm not convinced that the comparisons are sufficient.
* Higher computational cost for SPI (and lack of discussion/ acknowledgment), and lack of comparisons to methods with similar computational cost (15g5) -- I think this is the most crucial reason for the suggestion of rejection.